# Clinical Application of Epithelial Sodium Channel (ENaC) as a Biomarker for Arterial Hypertension

**DOI:** 10.3390/bios12100806

**Published:** 2022-09-29

**Authors:** Diana García-Rubio, Ivette Martínez-Vieyra, Maria Beatriz de la Mora, Marco Antonio Fuentes-García, Doris Cerecedo

**Affiliations:** 1Laboratorio de Hematobiología, Escuela Nacional de Medicina y Homeopatía, Instituto Politécnico Nacional, Mexico City 07738, Mexico; 2CONACyT Fellow-ICAT, Universidad Nacional Autónoma de México, Mexico City 04510, Mexico; 3Unidad de Investigación, Instituto Mexicano el Seguro Social, Mexico City 06700, Mexico

**Keywords:** platelets, gold nanoparticles, bioconjugates, hypertension diagnosis

## Abstract

Arterial hypertension (HTN) is a global public health concern and an important risk factor for cardiovascular diseases and renal failure. We previously reported overexpression of ENaC on the plasma membrane of human platelets is a hallmark of HTN. In this double-blinded study of an open population (*n* = 167), we evaluated the sensitivity and specificity of a diagnostic assay based on gold nanoparticles (AuNPs) conjugated to an antibody against epithelial sodium channel (ENaC) expressed on platelets, which is detected using a fluorescent anti-ENaC secondary antibody and spectrofluorometry. Using the cutoff value for the AuNP-anti-ENaC assay, we confirmed the diagnosis for 62.1% of patients with clinical HTN and detected 59.7% of patients had previously undiagnosed HTN. Although some shortcomings in terms of accurately discriminating healthy individuals and patients with HTN still need to be resolved, we propose this AuNP-anti-ENaC assay could be used for initial screening and early diagnosis to critically improve opportune clinical management of HTN.

## 1. Introduction

The emerging epidemic of arterial hypertension (HTN) represents a global public health concern as HTN is related to several kidney and cardiovascular system complications. The prevalence of HTN is 40% in México, and the incidence of HTN is higher when associated with obesity (42.3%) or *diabetes* (65.6%). HTN is a disease that lacks early symptomatology; in addition, arterial tension values can be altered by multiple unrelated causes, which hinders early detection and diagnosis and may explain the high percentage of individuals (~40%) with undiagnosed HTN [1].

HTN is a heterogeneous disease, and several factors, including smoking, aging, hypercholesterolemia, hyperglycemia, obesity, elevated C-reactive protein, and chronic systemic infection, have been identified to generate high blood pressure as they promote oxidative stress and inflammatory vascular responses [2,3]. Therefore, candidate biomarkers for hypertension should identify oxidative stress and inflammation; adipokines were recently proposed as potential molecules to detect HTN [4].

The quantum effects and surface area of nanomaterials confer unique potential to detect analytes of medical interest, as the nanomaterials can be measured based on either their electrochemical [5] or optical responses [6]. In particular, silver and gold nanoparticles, which exhibit localized surface plasmon resonance (LSPR), are excellent candidates for optical biosensors [7]. The LSPR effect allows colorimetric changes, agglutination tests, and absorption band displacement to be used as indicators of the amount of the analyte of interest for a variety of biomarkers [8]. In addition, metal nanoparticles can be employed to increase the sensitivity of detection techniques such as immunofluorescence [9] or Raman spectroscopy [10]; therefore, there is an increasing number of research studies, prototypes, and novel technologies focused on the development of diagnostic methods [11,12].

The Epithelial Sodium Channel/Degenerin (ENaC/DEG) family belongs to a superfamily of sodium-selective channels composed of three different subunits, namely, α, β and γ-ENaC, encoded by three different genes (SCNN1A, SCNN1B and SCNN1G, respectively) [13]. ENaC is expressed in the distal colon, ducts of exocrine glands, lungs, and apical surface of the epithelial surface of the distal nephron [14], where it is responsible for the reabsorption of sodium and regulates the extracellular fluid volume and consequently blood pressure [15].

Liddle syndrome is a congenital disorder due to a single gene mutation in β and γ-ENaC subunits that impairs their degradation. Therefore, the quantity of these channels on the apical surface of the distal nephron increases [16], promoting an increase in sodium reabsorption and chronic volume retention generating a hypertensive state [17,18].

Based on this evidence and due to the wide presence of ENaC in different cell types including blood cells, we previously identified and validated that epithelial sodium channel (ENaC) is overexpressed on the membrane of platelets of patients with HTN regardless the time of evolution, medication or anthropometric characteristics of patients and proposed that relative quantification of ENaC could be employed as a biomarker to diagnose HTN [19]. To improve the sensitivity of this approach for the diagnosis of HTN, we functionalized gold nanoparticles to facilitate the coupling of an antibody raised against ENaC (AuNP-anti-ENaC) expressed on platelets. We showed that the amount of ENaC expressed on the platelets of patients with HTN could be quantified using a secondary antibody labeled with a fluorochrome, and the levels of fluorescence detected were proportional to the expression of ENaC on platelets [20].

Here, we present a double-blinded clinical study to evaluate the clinical utility of the ENaC fluorescence immunoassay on a larger scale among 167 subjects attending the Unidad de Medicina Familiar No. 44 del Instituto Mexicano del Seguro Social in Mexico City. Platelet pellets were collected from individuals receiving clinical consultations or undergoing clinical laboratory tests, regardless of their arterial tension values or hypertension diagnosis. The sensitivity and specificity values determined using receiving operating curves (ROC) suggest our proposed test could be implemented in clinical practice, especially to identify the large proportion of individuals with undiagnosed hypertension. Clinical aspects and anthropometric characteristics related to hypertension are also discussed, as well as the basic aspects associated with the relationship between the gold nanoparticles and fluorophore and a possible quenching effect. Finally, we compared the effectiveness of the immunoassay with Western blotting, which is a proven method used to quantify proteins such as ENaC. The advantages and disadvantages of each technique are discussed in the context of the implementation of massive-scale screening for HTN using ENaC as a biomarker.

## 2. Materials and Methods

### 2.1. Platelet Samples

Blood samples were obtained voluntarily from individuals undergoing routine clinical analyses (regardless of their arterial tension values) in October and November 2020, at the clinical laboratory of Unidad de Medicina Familiar No. 44 IMSS. Approval for this study was given by the National Committee of Instituto Mexicano del Seguro Social (IMSS) R-2020-785-150. Samples were coded as “MV” followed by a number, and the coding was stored separately to the patients’ personal information. The anthropometric data for the patient cohort are summarized in Table 1.

Peripheral venous blood was drawn from the antecubital vein into 4.5 mL plastic whole blood collection tubes spray-coated with K_2_EDTA (Vacutainer; BD, Franklin Lakes, NJ, USA) and processed within 30 min. Platelet-rich plasma was obtained by centrifugation at room temperature; then, the plasma was mixed with an equal volume of citrate anticoagulant and centrifuged at 400× *g* for 10 min. The washed platelet pack was suspended in Hank’s balanced saline solution (HBSS) deprived of calcium (137 mM NaCl, 5.3 mM KCl, 1 mM MgCl_2_, 0.28 mM Na_2_HPO_4_.12H_2_O, 0.87 mM NaH_2_PO_4_, 0.44 mM KH_2_PO_4_, 4.1 mM NaHCO_3_, and 5.5 mM glucose) for all assays.

### 2.2. Calibration Curves

Five serial dilutions of recombinant ENaC (Cat No. APREST71796; Sigma Chemical Co., St. Louis, MO, USA) were prepared in triplicate to obtain concentrations of 10, 20, 30, 40, and 60 µg/mL, and then processed by Western blotting. Negative controls without secondary antibody were included. It is important to mention that a calibration curve could not be generated for the AuNP-anti-ENaC assay using the recombinant ENaC, as the antibody does not bind to ENaC in lysed platelets.

### 2.3. Western Blotting

Resting platelets (1 × 10^6^) were lysed by boiling in sodium dodecyl sulfate (SDS) and β-mercaptoethanol for 5 min, subjected to 10% SDS-polyacrylamide gel electrophoresis (PAGE), and transferred onto nitrocellulose membranes using a semidry system (Thermo Electron Co., Milford, MA, USA). Membranes were incubated with α-ENaC rabbit polyclonal antibody (1:200; catalog: TA329045; OriGene Technologies Inc., Rockville, MD, USA) or an anti-mouse GAPDH antibody (1:500; cat. sc-47724; Santa Cruz Biotechnology, Inc., Santa Cruz, CA, USA) as a loading control at 4 °C overnight. The membranes were then washed and incubated with horseradish peroxidase-conjugated anti-rabbit (1:4000; cat No. 2357; Santa Cruz Biotechnology, Inc, Santa Cruz CA, USA) or anti-mouse secondary antibodies (1:6000; cat No. A9917; Sigma-Aldrich). After washing with TBS/T three times, enhanced chemiluminescence (ECL) assays were performed to visualize the bands on X-ray film. Negative control blots were incubated solely with the horseradish peroxidase (HRP)-conjugated secondary antibodies. Densitometry analysis was performed by two different technicians, using Win Image Studio Digits Ver 4.0 software (LI-COR, Inc., Lincoln, NE, USA); the signal intensities observed for each individual were normalized to the respective signal of the housekeeping protein (GAPDH). This process was performed twice for each individual included in the study.

### 2.4. Preparation and Characterization of AuNP-anti-ENaC Conjugates

Functionalization of gold nanoparticles (AuNPs) was conducted as previously described by García-Rubio et al. (2020) [20]. In brief, 4.5 × 10^10^ gold nanoparticles/mL (Cat. No. 753629; Sigma-Aldrich Mexico, Mexico) were mixed with 80 μM of 3-mercaptopropionic acid (MPA) in a water in a 1:1 vol ratio and sonicated continuously for ~1 h to completely functionalize the nanoparticles. Then, carboxylic group activation and AuNP-anti-ENaC conjugation were induced by the addition of 24 mM NHS esters (N-hydroxysuccinimide esters) and 12 mM EDC (1-ethyl-3-(-3-dimethylaminopropyl) carbodiimide hydrochloride) and 12 mM EDC to a suspension of AuNPs with a concentration of 0.25 mM. The mixture was stirred at room temperature for 10 min. The functionalized AuNPs were incubated with α-ENaC, a rabbit polyclonal antibody (Cat. No.: TA329045; OriGene Technologies Inc., Rockville, MD, USA) diluted in PBS containing 0.3 mg/mL BSA, with stirring for 15 min at room temperature. Finally, the conjugates were washed and separated by centrifugation and resuspended in PBS.

Functionalized AuNPs and AuNP-anti-ENaC conjugates were characterized by FT-IR spectroscopy using an ATR (Smart-iTX; Thermo-Fisher, Waltham, MA, USA) accessory coupled to a FTIR spectrometer (Nicolet 912A0712 iS5; Thermo-Fisher, Waltham, MA, USA). UV–VIS measurements were carried out in quartz cuvettes on a Cary 5000 UV–VIS NIR spectrometer (Thermo-Fisher, Mexico, Mexico), and the particle sizes were determined by Zeta potential measurement via dynamic light scattering (DLS; Zetasizer Nano-ZS90, Malvern Instruments, Malvern, UK).

### 2.5. Biosensing of ENaC in Platelets Using a Fluorescent Assay

AuNP-anti-ENaC conjugates were incubated for 1 h at room temperature with 2.2 × 10^6^ platelets from each individual, then incubated with secondary antibody labeled with Alexa Fluor 488 fluorophore (λ excitation: 488 nm, λ max emission: 520 nm) for 1 h at room temperature; then, the platelets were washed twice with PBS and separated by centrifugation at 500× *g* for 3 min. The fluorescence emission of the bioconjugates in PBS was measured in quartz cells at an excitation wavelength of 490 nm.

### 2.6. Sample Size Calculation

The aim of the present study was to validate the AuNP-anti-ENaC–platelets assay as a diagnostic tool with a sensitivity of 90% and specificity of 90%. As the prevalence of HTN in the Mexican population is 40% (ENSANUT 2016 MC) [1], the sample size was calculated as [21]:TP + FN = Z^2^ × Sensitivity (1 − Sensitivity)/W^2^
TN + FP = Z^2^ × Specificity (1 − Specificity)/W^2^
where: TP, true positives; FN, false negatives; TN, true negatives; FP, false positives; Z, normal distribution value (1.96); W, maximum acceptable width of the 95% confidence interval set to 10%.
TP + FN = 34.5
TN + FP = 34.5

The N needed to achieve a sensitivity of 90%, and the N needed to achieve a specificity of 90% and prevalence (P) of 40% were calculated as follows:

For sensitivity:TP + FN/P = 34.5/0.4 = 87 patients with HTN

For specificity:TP + FN/1 − P = 34.5/0.6 = 58 normotensive individuals 

The total sample size required was 146 participants; however, we included a total of 167 individuals.

### 2.7. Electron Microscopy

Suspended AuNP-anti-ENaC conjugates were incubated with 2.2 × 10^6^ platelets and then fixed in 2.5% glutaraldehyde/formaldehyde in Dulbecco’s phosphate-buffered saline (DPBS) with a pH of 7.4 for 30 min. Samples were settled on silver grids and observed using a Jeol 1400 transmission electron microscope (JEOL, Tokyo, Japan) at 80 keV and the scanning electron microscopy JSM-7800F (JEOL, Tokyo, Japan).

### 2.8. Blinding and Statistical Methods

One investigator attended the clinic and enrolled the patients for this study and coded the samples. Other investigators performed the assays and analyzed the data. The samples were only uncoded after the data analysis was complete. Statistical analysis was conducted using GraphPad Prism 6 software (La Jolla, CA, USA). All reported 𝑃-values are two-sided, with *p* < 0.05 being considered statistically significant.

### 2.9. Principal Components Analysis (PCA)

PCA was used to identify a smaller number of uncorrelated variables that are easier to interpret and analyze than the observed variables measured. To identify the samples that are similar or different to one another to find out which variables make one group different from another. PCA was performed using RStudio software ver. 2022.07.1 + 554 (RStudio, Inc. Boston, MA, USA).

## 3. Results

### 3.1. Bioconjugates of AuNPs

Bioconjugation was achieved by forming a peptide bond between the N-terminal amino group of the antibodies and the carboxylic groups exposed on the surface of the AuNPs after functionalization with MPA. The characteristic band of amide or peptide can be observed in the FTIR-ATR spectrum (Appendix A). The band at 1650 cm^−1^ corresponds to the amide bond [22] due to the presence of albumin and the anti-ENaC antibody used during the synthesis of the bioconjugates. Albumin is used for effective reduction of nonspecific binding [23].

The zeta potential values and size distribution of the AuNPs measured by DLS are shown in Appendix A. We determined that 0.3 mg/mL BSA was the optimal concentration for the AuNP-anti-ENaC conjugation procedure, as the resulting NPs had the narrowest size distribution, and their zeta potential value was close to that of unmodified nanoparticles (−36.7 ± 1.6 mV). Moreover, a 10 nm-thick BSA/anti-ENaC protein-antibody complex surrounds the NPs, creating working complexes with a total diameter of 40 nm. Details of the conjugation process can be found in García-Rubio et al. [20].

The AuNP colloidal solution exhibits a distinctive red color and an absorption band in the visible range related with plasmonic behavior. UV–VIS spectra show the absorption band centered at 524 nm (Appendix A); the intensity of the band decreased in intensity after functionalization with mercaptopropionic acid (AMP), as described before [20], and was slightly red-shifted. When the ENaC antibody was attached to the AuNPs by an amide bond, the absorption band became broader and red shifted to 532 nm, indicating an association between anti-ENaC and the surface of the AuNPs (Appendix A). The absorption band is highly sensitive to the surroundings of the nanoparticles surface, and thus, the presence of AMP, anti-ENaC antibody, and albumin leads to a notorious modification of the width and the intensity of the absorption band. The modification in the spectra along with the FTIR-ATR spectroscopy can be related to the successful functionalization of the AuNP.

Scanning electron microscopy (SEM) images allowed to measure the size of the AuNPs and their increase in this characteristic, which they acquire after functionalization, when they are associated with albumin and anti-ENaC (Appendix A). The presence of albumin and anti-ENaC in functionalized AuNPs was also clearly appreciated with the transmission electron microscopy (TEM) (Appendix A).

### 3.2. Patient Characteristics

Based on arterial tension values measured using a validated oscillometric semiautomatic sphygmomanometer at the time of taking the blood sample, the 167 individuals in this study were classified as normotensive (normotensive individual group [NI] *n* = 39) or as having elevated arterial tension with HTN (*n* = 128). We adopted the blood pressure values of the American College of Cardiology/American Heart Association Hypertension Guideline 2017; therefore, systolic and diastolic values < 120/80 mm Hg were considered normal; elevated blood pressure was defined as 120–129/<80 mm Hg, while HTN was considered as ≥140/90 mm Hg [24].

The patients were aged between 25 and 90 years old. The median age of the normotensive individual group (NI) was significantly lower (44 years) than that of the patients with diagnosed HTN (DHTN; 60 years; *p* < 0.0001) or previously not known to be hypertensive (NKHTN; 56 years; *p* = 0.008; Table 1). The sex distribution of the NI and DHTN groups was similar (*p* = 0.743 and *p* = 0.821, respectively); however, the NKHTN group contained significantly more women than men (*p* = 0.0382/0.069).

The DHTN and KNHTN groups had significantly higher median systolic values (137 mm Hg; *p* < 0.001) compared to the NI group (115 mm Hg); while the diastolic values of the DHTN and KNHTN groups were significantly higher than that of the NI group (84 vs. 70 mm Hg; *p* < 0.001 and 89 vs. 70 mm Hg, *p* = 0.001 respectively; Table 1).

In relation to comorbidities, *diabetes* was significantly more common (71%) in the DHTN group than the NI group (31.3%; *p* = 0.001) or NKHTN groups (38.4%; *p* = 0.767). Dyslipidemia was significantly more common (71%; *p* = 0.008) in the DHTN group compared to the NI group (25.4%) and NKHTN group compared to the NI group (19.4%; *p* = 0.667). The NI group had the lowest mean body mass index (BMI; 21.5 kg/m^2^), while the mean BMI values of the DHTN and NKHTN groups were classified as overweight (29.2 kg/m^2^ and 30.6 kg/m^2^, respectively; Table 1).

Current consumption of alcohol, cigarettes, or recreational drugs may also contribute to the development of HTN. The DHTN group consumed alcohol, cigarettes, and recreational drugs (60.9%, 34.8%, and 7.3%, respectively) significantly more frequently than the NI group (24%, 20%, and 5.3%, respectively; *p* = 0.017). Similarly, the individuals in the DHTN group more frequently reported practicing exercise than the NI and NKHTN groups (63.8%, 22.4%, and 29%, respectively; Table 1).

Antihypertensive agents were only used by individuals with previously diagnosed HTN (Table 2); most of these patients (80.3%; *n* = 53/66) were using monotherapy, while 19.7% (*n* = 13/66) used multiple antihypertensive agents. Angiotensin receptor blockers (ARB) were the most commonly prescribed type of medication (37.9%), followed by angiotensin II receptor blockers (ACEI; 22.8%), beta blockers (BB; 10.6%), and calcium channel blockers (CCB; 9%); 19.7% of the patients in the DHTN group were using a combination more than one of these four kinds of medication (Table 2).

### 3.3. Principal Component Analysis (PCA)

PCA was conducted on the systolic, diastolic, diabetic, dyslipidemia, obesity, and age for each patient. PCA reduced the total number of important anthropometric characteristics into fewer uncorrelated orthogonal principal components (PCs).

Only orthogonal PCs were retained if the eigenvalues were equal to or greater than one. Therefore, the first retained PC (PC-1) explained the largest variance, with all subsequent PCs (e.g., PC-2) explaining a smaller proportion of the total variance that was not accounted for in the previous PC. To visually compare the size of the eigenvalues, we used the bar plot (Figure 1).

To assess the data structure and detect clusters, outliers, and trends, we grouped the data in the score plot (Figure 2). The PCA plot obtained shows the two principal components or PCs (PC1 + PC2) cumulatively explain 71.8% of the variance present in the samples.

According to the vectors displayed in the plot, it is evident that systolic and diastolic arterial tension are positively correlated, as well as dyslipidemia and *diabetes*; contrary to expectations, obesity and systolic and diastolic arterial tension are not likely correlated with *diabetes* and dyslipidemia. DHTN, NKHTN, and HI (left bottom quadrant) showed a negative correlation with dyslipidemia and *diabetes*, whereas DHTN, NKHTN, and a few HI (right upper quadrant) are independent of dyslipidemia and *diabetes* condition. Only some DHTN and KNHTN individuals (left upper quadrant) are directly related with their systolic and diastolic tension values. Most of HI (right bottom quadrant) had a negative correlation with arterial tension values.

The PCA models generated from patients with or without comorbidities indicated that both *diabetes* and dyslipidemia were important indicators responsible for the separation between DHTN, NKHTN, and HI. In contrast, most of HI were located in the lower right quadrant.

According to the biplot used, we determined that dyslipidemia and *diabetes* have large negative loadings on component 2; this component focuses on comorbidities of hypertension. The points in the lower right-hand corner may be outliers.

We used the outlier plot to identify the feasible points that could be above the reference line that might significantly affect the results of the analysis. We identified 2 outliers that corresponded to patient 21 and 158 and a confidence level of 0.975 (Figure 3).

### 3.4. Accuracy of ENaC as a Biomarker for HTN

Previously, we proposed ENaC as a biomarker to identify patients with HTN [19] and developed a biosensor using AuNPs to detect ENaC expressed on platelets [20]. In order to validate ENaC as a biomarker for HTN, we first established a calibration curve using a range of concentrations (10 µg/mL to 60 µg/mL) of commercially available pure ENaC.

The calibration plot shown in Figure 4 revealed a robust, linear relationship between the ENaC concentration and the absorbance values detected by Western blotting (R^2^ = 0.9942). The limit of detection (LOD) was determined to be 0.035 µg/mL through the 3σ rule (three standard deviations from the mean) [25]. The linearity and low LOD indicate that detection of ENaC by Western blotting could be applied to accurately and sensitively quantify the physiological concentration of ENaC. These results allow us to include the Wb technique as an adequate reference to be compared later with the AuNP-anti-ENaC assay.

The limit of detection corresponding to the AuNPs-anti-ENaC assays could not be performed, since the commercially acquired ENaC (recombinant ENaC) or the one found in platelet lysates does not bind to the bioconjugates.

A calibration curve was generated using a range of ENaC concentrations (10 to 60 µg/mL) and resolved by Western blotting (Wb); the band intensities were determined in arbitrary units (A.U.). Each point on the graph is the mean of three replicates.

### 3.5. Expression of ENaC on Platelet Samples from Patients with Hypertension

As ENaC represents a continual variable, we performed receiver operating characteristic (ROC) curve analysis—the most widespread methodology used in biomarker validation—to identify a cutoff point for this continual variable. It is assumed that for the given cutoff point, *c*, in the continuum of the biomarker measurement, any measurement greater than *c* will be indicative of disease. The false positive fraction (FPF) or else 1-specificity is considered as indicative of a non-diseased individual, while the sensitivity or true positive fraction (TPF) is the proportion of times the biomarker identifies a diseased individual [26]. Therefore, the FPF included patients for whom their TA values were considered NT according to the values defined by the American College of Cardiology/American Heart Association Hypertension Guideline 2017 [24]; for the TPF, we only included patients previously diagnosed with hypertension by the physician.

Both the AuNP-anti-ENaC assay and Western blotting were used to determine the relative ENaC expression levels in platelets in the patient cohort, which included patients with hypertension (DHTN) and normotensive individuals (NI).

We calculated the AUC values for our proposed biomarker (ENaC) for both Western blotting and the AuNP-anti-ENaC assay.

For Western blotting, the AUC corresponded to 0.8331, with a sensitivity of 0.7509% (CI: 0.7405–0.9256), specificity of 0.7090% (CI: 0.4779–0.8087), LR of 2.558, and cutoff value of 0.8906. The AUC obtained from the spectrofluorometer readings of the AuNP-anti-ENaC assay was 0.7303, with a sensitivity of 0.7472% (CI: 0.5841–0.8067), specificity of 0.5849% (CI: 0.5531–0.7942), LR of 2.218, and cutoff of 0.5219 (Table 3).

Based on the data above, we examined the accuracy of ENaC as a biomarker for hypertension in the Western blotting and AuNPs assays by calculating the Youden index (*J*-index) [27]:Youden index = sensitivity + (specificity − 1)

The *J*-index for Western blotting was 0.4591, while the *J*-index for the AuNP-anti-ENaC assay was 0.3341. *J* represents the point of the ROC curve farthest from chance, and it exists at the point in which equal weight is given to sensitivity and specificity.

The accuracy of the test refers to its capability to measure the true amount or concentration of an analyte for which it was created in a sample.

The accuracy for the tests was determined using the following formula:Accuracy = True Positive + True Negative/True Positive + True Negative + False Positive + False Negative.

The accuracy for Western blotting was 0.7388, and for the AuNP-anti-ENaC assay, it was 0.6726.

In Figure 5Aa, we show the representative bands of 55 kDa corresponding to α-ENaC from five different individuals, as well as the respective GAPDH bands used to normalize the values. Each lane was loaded with 50 µg/mL of total platelet protein. The densitometry analysis was performed for each sample. The intensities of the bands and the results of these assays are summarized in Figure 5Ab.

Expression levels of a-ENaC detected by Western blotting assays were significantly increased in blood samples of patients with HTN compared to controls (mean = 0.760 vs. 1.28; *p* < 0.0001) Figure 5Ab. The AUC value of the ROC curve for α-ENaC relative expression detected by Western blotting was 0.8331, with a standard error of 0.04721, and 95% confidence interval 0.7405–0.9256, indicating a good accuracy of the biomarker (Figure 5Ac).

In Figure 5Ba, we show representative fluorescence spectra of different individuals; the intensity of fluorescence of the samples allowed to separate them into two groups. The maximum fluorescence values for each sample were used to distinguish between patients with hypertension and healthy individuals. Expression levels of α-ENaC detected by the AuNPs assay was significantly increased in blood samples of patients with HTN compared to controls (mean = 0.476 vs. 0.678; *p* < 0.0001) (Figure 5Bb). The AUC values of the ROC curve for ENaC relative expression detected by AuNPs-anti-ENaC assay was 0.7303, standard error 0.0317, 95% confidence interval 0.6456–0.8149 and *p* value < 0.0001 indicating a moderate accuracy of the method (Figure 5Bc).

### 3.6. Identification of Patients with Undiagnosed Hypertension Using the Proposed Diagnostic Assay

Next, we determined if the cutoff value of the AuNP-anti-ENaC assay was useful to confirm the diagnoses of patients with HTN and also to identify individuals with possible undiagnosed HTN. We compared the relative expression of ENaC on platelet membranes determined by the AuNP-anti-ENaC assay for patients that registered high arterial tension values. Using the calculated cutoff values for the AuNP-anti-ENaC assay, we confirmed the diagnosis of HTN for 62.1% (41/66) of the DHTN group. Furthermore, the AuNP-anti-ENaC assay suggested a diagnosis of HTN for 61.3% (38/62) of the individuals in the NKHTN group. The Western blotting assay confirmed the diagnosis of HTN for 68.2% (45/66) of the patients in the DHTN group and suggested a diagnosis of HTN for 59.7% (37/62) of the individuals in the NKHTN group.

### 3.7. Topographical Distribution of AuNP-anti-ENaC Bioconjugates on Platelets from Patients with HTN

To visualize the distribution of AuNP bioconjugates on the plasma membrane, platelets from patients with HTN were dropped onto cupper grids and observed via transmission electron microscopy (TEM). The images revealed a lower density of bioconjugates on platelets from individuals in the NT group (Figure 6A) compared to the higher density of bioconjugates attached to the plasma membrane of platelets from patients with HTN (Figure 6B). In both cases, the bioconjugates were observed as electrodense spots homogeneously distributed at the plasma membrane.

## 4. Discussion

HTN has a high prevalence and represents the main risk factor for cerebral and cardiovascular events; however, the disease lacks specific symptoms and is frequently undiagnosed. Although some blood biomarkers have helped to improve our understanding of the pathogenesis, diagnosis, progression, and therapeutic efficacy, and even helped to determine the prevalence of hypertension [28], these biomarkers are not considered in clinical laboratories and a high percentage of patients are unaware that they suffer from hypertension. For our knowledge, this is the first time that the expression of ENaC on the plasma membrane of platelets has been proposed as a biomarker to detect HTN [19]; in addition, ENaC is not related to oxidative stress or inflammatory processes. To improve the sensitivity of this approach for the diagnosis of HTN, we coupled functionalized gold nanoparticles to an antibody raised against ENaC (AuNP-anti-ENaC) expressed on platelets. The relative levels of ENaC on the platelets are determined using a secondary antibody labeled with a fluorochrome [20]. As clinical studies are necessary to scale-up this gold nanoparticle-based immunoassay to a well-established diagnostic test for HTN, we evaluated the accuracy of the AuNP-anti-ENaC assay using a double-blinded study of platelet samples from an open population (*n* = 167) that excluded pregnant women.

To validate our proposed biomarker (AuNP-anti-ENaC), we plotted ROC curves to characterize the measurements arising from all of the subjects included in the study. The AUC of a ROC curve reflects the amount of separation of the distribution of the biomarker between the non-diseased and diseased populations [29]. Although there are no absolute rules in terms of how large the AUC must be to confirm the predictive value of an assay, it is assumed that AUCs of 0.75 are considered medically useful. Our AuNP-anti-ENaC assay exhibited an AUC of 0.7303, with a sensitivity of 0.7472% (CI: 0.64565–0.8149) and specificity of 0.5849% (CI: 0.5531–0.7942; Table 3). Even though these values are not far from the assumed clinically useful values, we are aware that further studies including a higher number of subjects are required to achieve a higher AUC value and narrower confidence interval.

The International Society of Hypertension (ISH) and related associations such as American Heart Association (AHA), the European Society of Cardiology (ESC), and the European Society of Hypertension (ESH) [30] have developed worldwide practice guidelines for the diagnosis of hypertension in adults and established that measurements of arterial tension must be performed with a cuff oscillometric sphygmomanometer which is the only way to diagnose HTN.There are optimal conditions that must be taken into account for blood pressure measurement in the doctor’s office and at home before making a diagnosis; unfortunately, they are left out of consideration, even by specialists [31]. The main motivation to develop a diagnostic tool for this disease was to overcome all the situations that prevent an opportune diagnosis of HTN.

The prevalence of non-cardiovascular comorbidities (*diabetes* or dyslipidemia) was significantly higher among the DHTN and NKHTN groups of patients with hypertension than in the NI group (*p* = 0.001 and 0.008, respectively; Table 1). Obesity has become an important health problem with a high prevalence among patients with HTN and individuals with undiagnosed HTN; hence, timely diagnosis of HTN has become even more important. According to our PCA results, patients with *diabetes* and dyslipidemia should be monitored for hypertension and seek lifestyle changes. However, it is important to mention that PCA points to the variability between individuals and that associated diseases such as dyslipidemia or *diabetes* do not necessarily promote the development of hypertension. In relation to the two patients that were identified as outliers, we considered them as part of the variability inherent to biological systems, we did not exclude them from the study as we did not consider them irrelevant for the analysis. Overall, our findings support the need for a diagnostic test for hypertension independent of patient characteristics and associated diseases.

We considered 59.7% (37/62) of individuals with no previous diagnosis of HTN to be hypertensive based on the AuNP-anti-ENaC assay, and these patients exhibited high arterial tension values at the moment we took their blood sample in the clinic. It has been accepted that detection of raised blood pressure is more accurate when monitored at home compared to measurements in the clinic [32]. Additionally, it is known that blood pressure varies from person to person and is susceptible to fluctuations due to the complex interplay between several cardiovascular regulatory mechanisms, physical activity, environmental conditions, and circadian rhythm [33].

The evidence we have using ENaC as a biomarker of HTN in different groups of individuals indicates that the levels of ENaC appear to remain constant and are independent of external factors that might alter the diagnosis based on blood pressure alone. To ensure the reproducibility of the optical response, which depends on the size and polydispersity of the nanoparticles, they were purchased from Sigma-Aldrich (size ~30 nm and Polydispersity Index ≤ 0.2). Functionalization with mercaptopropionic acid at the selected concentration maintained in all cases (100% of the times prepared) the colloidal dispersion of the nanoparticles showing absolute zeta potential values greater than 37 mV; and regarding the conjugation of the antibodies and the nanoparticles, specific concentrations of anti-ENaC and albumin were added and covalently bound to the carboxyl-terminated nanoparticles. This gives the bioconjugates sufficiently reproducible interaction properties, since the excess of protein added helped to completely block any possible non-specific interaction at potential binding sites on the surface of the bioconjugates when they interact with platelets.

Some epidemiologic studies have stated that women have a lower prevalence of HTN than men; however, we found similar numbers of both genders had HTN. This might be due to the fact the population analyzed was homogeneous in terms of race, ethnicity, and obesity [34]. In contrast, there were more women than men between the fourth and fifth decades of life in the NKHTN group than in the NI group. Thus, the hormonal component could be an important factor in the development of HTN [35]. Older adults have been reported to be more dedicated to taking care of their health and more frequently identify asymptomatic diseases. Age is a worldwide, natural factor that increases arterial tension and increases the risk of HTN [1,36,37].

As this study had a descriptive cross-sectional design, we could not evaluate the effects of dietary changes or exercise on the progression of the disease or the clinical diagnosis of individuals with undiagnosed HTN; we recognize these points as limitations of the present study. However, using our proposed assay, we achieved the goal of our study of detecting an important percentage of individuals with undiagnosed HTN, which makes it possible to suggest close monitoring of these individuals to confirm their diagnosis of HTN and begin timely treatment.

In patients with HTN, platelets circulate in an activated state characterized by loss of the typical discoid shape to cells with numerous membrane projections [19]. This morphological change may facilitate closer interactions between the AuNP-anti-ENaC complexes on the plasma membrane of platelets. On the other hand, quenching has previously been observed to occur when two fluorescent particles are separated by a distance of less than 20 nm [38]. The transmission electron microscopy photographs confirmed the presence of bioconjugates on the plasma membranes of the analyzed platelets. The abundance of bioconjugates was lower on platelets from normotensive individuals (Figure 6A) than on platelets from patients with HTN, in which the closeness of the AuNP-anti-ENaC at the plasma membrane was evident (Figure 6B). Thus, a quenching phenomenon might possibly explain the lower intensity of fluorescence detected on platelets from some patients with HTN and, as a consequence, the relatively low sensitivity and specificity of the AuNPs assay. While the assay could be easily scaled up, this quenching issue is the main limitation of the proposed biosensor. To overcome this issue, we suggest the development of another AuNP-anti-ENaC assay in which the detection method is not based on fluorescent molecules.

## 5. Conclusions

Clinical trials are of major importance to the scale up of novel diagnostic techniques involving nanotechnology. Our careful double-blinded study of platelets from 167 subjects demonstrates the potential of an immunofluorescent bioconjugate assay based on gold nanoparticles to detect ENaC as a clinical diagnostic biomarker for HTN. The moderate sensitivity and specificity of our AuNP-anti-ENaC-platelets assay could possibly be related to quenching due to changes in the distance between the gold nanoparticles and fluorophore induced by constitutively activated platelets in patients with HTN; analytical modifications to control the distance between the nanoparticles may help to overcome these issues. Further research is necessary to improve the sensitivity and scale up this assay for routine clinical use as a screening technique for the early detection of HTN, which may help to improve patients’ quality of life and prognosis.

## Figures and Tables

**Figure 1 biosensors-12-00806-f001:**
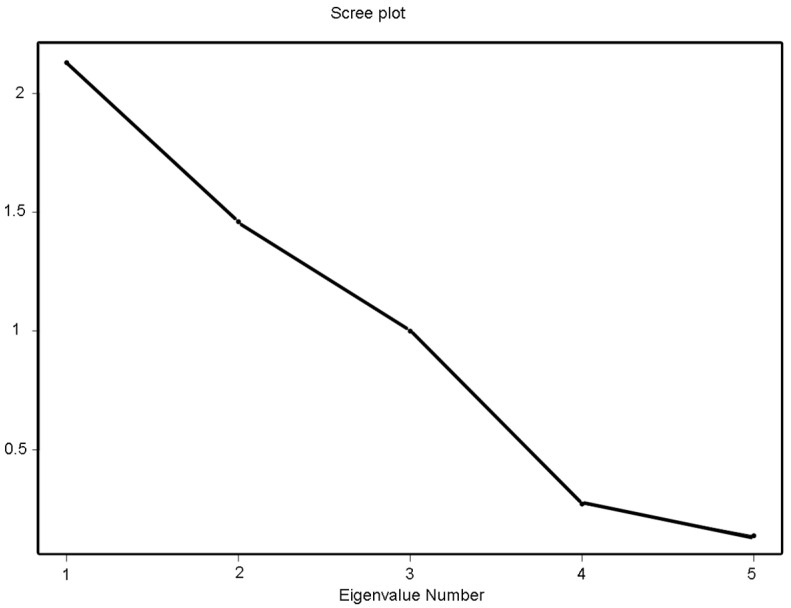
Scree plot shows the first three principal components have eigenvalues greater than 1. The three first components explain 85% of the variation in the data.

**Figure 2 biosensors-12-00806-f002:**
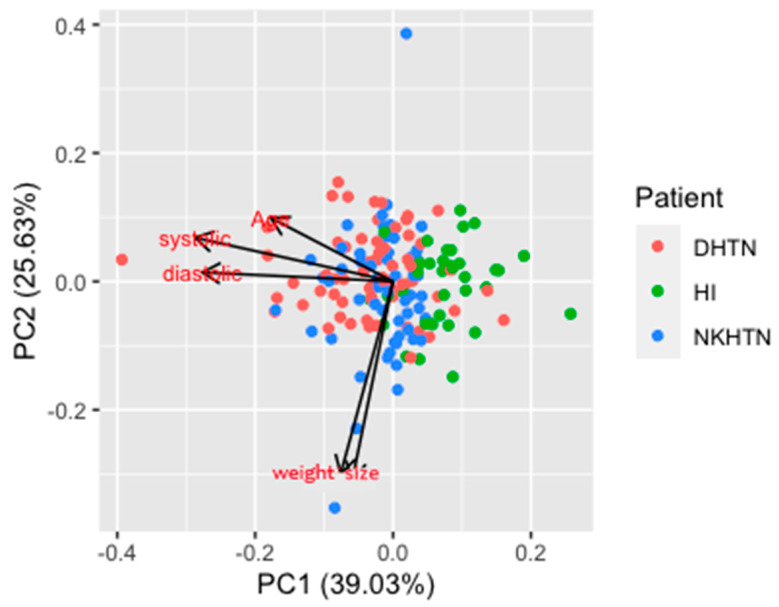
Loading plot. Loading plot including systolic and diastolic arterial tension, *diabetes*, dyslipidemia, obesity, and age.

**Figure 3 biosensors-12-00806-f003:**
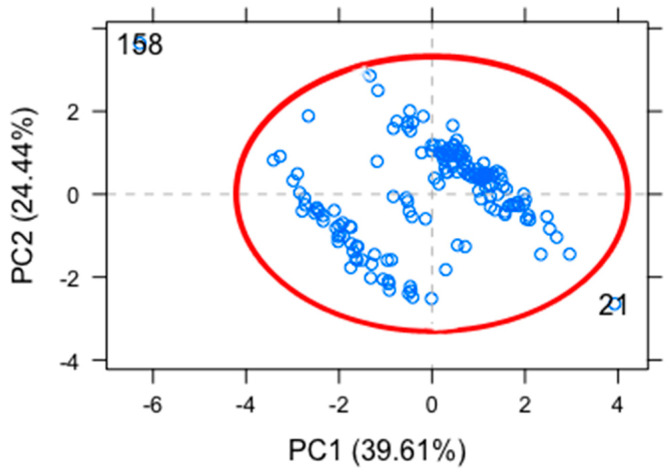
Outlier plot. The individual’s dataset included in the study, with sample 21 and 158 as outlier samples.

**Figure 4 biosensors-12-00806-f004:**
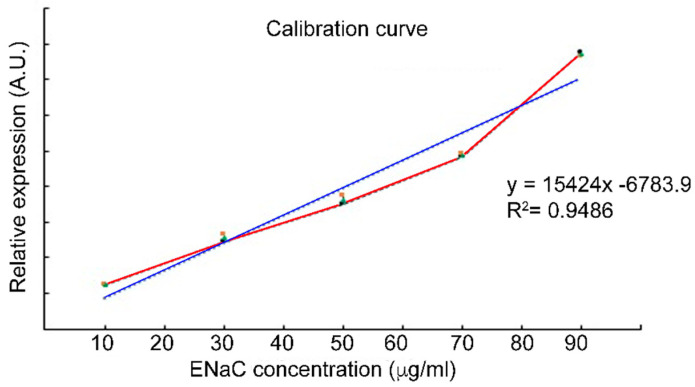
Epithelial sodium channel (ENaC) calibration curve. The red line represents the different ENaC concentrations. The blue line is the linear regression for the entire set of standard points.

**Figure 5 biosensors-12-00806-f005:**
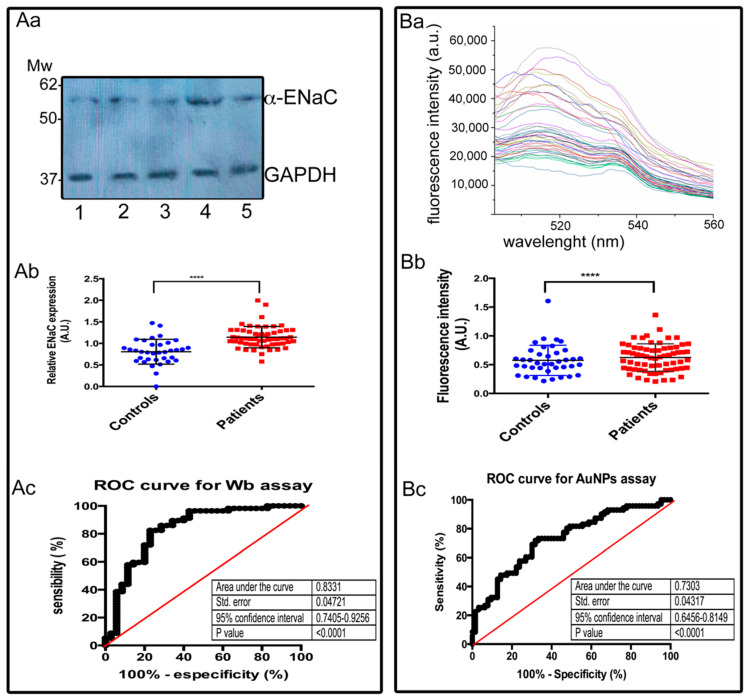
Epithelial sodium channel (ENaC) expression detected on platelet plasma membranes: (**Aa**) Representative bands corresponding to α-ENaC (55 kDa) and loading control GAPDH (37 kDa) detected by Western blotting assay from platelet lysates from individuals included in the study regardless of their health condition. (**Ab**) Levels of ENaC expression detected on platelet plasma membranes using the blotting assay. Band intensities are expressed as the relative number of pixels; **** *p* < 0.0001. (**Ac**) ROC curve. Black circles show the percentage of positive individuals for every cut-off value determined by Wb assay; red line represents the discrimination line, limiting the zone of no discrimination. (**Ba**) Representative fluorescence emission spectra of platelets samples λexc of 488 nm (495–570 nm). Samples corresponded to different individuals included in the study regardless of their health condition. (**Bb**) Levels of ENaC expression detected on platelet plasma membranes using the AuNPs-anti-ENaC assay. Band intensities are expressed as the relative number of pixels; **** *p* < 0.0001. (**Bc**) ROC curve. Black circles show the percentage of positive individuals for every cut-off value determined by AuNPs-anti-ENaC assay; red line represents the discrimination line, limiting the zone of no discrimination.

**Figure 6 biosensors-12-00806-f006:**
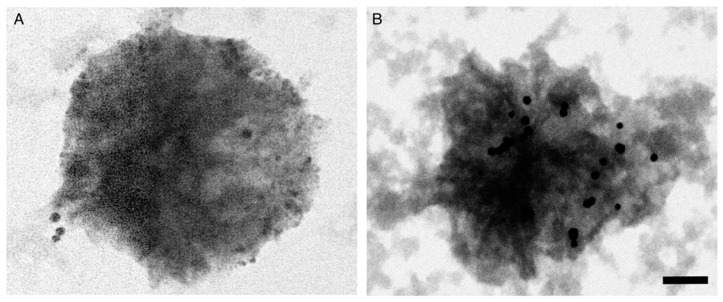
TEM images of bioconjugates and platelets. Topographical distribution of AuNP-anti-ENaC bioconjugates on the plasma membrane of platelets from (**A**) normotensive individuals and (**B**) patients with HTN. Scale bar = 0.5 µm.

**Table 1 biosensors-12-00806-t001:** Anthropometric characteristics of healthy individuals and patients with HTN.

	Normotensive Individuals (NI) (*N* = 39)	Diagnosed Hypertensive (DHTN) (*N* = 66)	Not Known to be Hypertensive (NKHTN) (*N* = 62)	*p* Value
Median age (IQR)	44 (32–90)	60.5 (50–66)	56 (47–61)	NI/DHTN = 0.0001 NI/NKHTN = 0.008
Women/men (%)	69.3/30.7	65.5/34.5	39/61	NI/DHTN = 0.743/0.821 NI/NKHTN = 0.0382/0.069
Systolic mmHg, mean (IQR)	115 (104–119)	137 (120–154)	137 (131–145)	NI/DHTN < 0.001 NI/NKHTN < 0.001
Diastolic mmHg, mean (IQR)	70 (67–79)	84 (77–93)	89 (84–95)	NI/DHTN < 0.001 NI/NKHTN < 0.001
Diabetic (%)	25.4	71	19.4	NI/DHTN = 0.001 NI/DHTN = 0.767
Dyslipidaemia (%)	31.3	66.7	38.7	NI/DHTN = 0.008 NI/NKHTN = 0.667
BMI (kg/m^2^) mean ± SD	21.5 ± 6.65	29.2 ± 5.62	30.6 ± 4.22	NI/DHTN = 0.1525 NI/NKHTN = 0.1318
Number of current smokers (%)	22.4	34.8	9.7	NI/DHTN = 0.413 NI/NKHTN = 0.619
Number of current alcohol consumption (%)	26.7	60.9	35.5	NI/DHTN = 0.017 NI/KNHTN = 0.617
Number of recreational drugs consumption (%)	6	7.3	---	NI/DHTN = 0.938
Number of practice exercise (%)	22.4	63.8	29	NI/DHTN = 0.006 NI/NKHTN = 0.716

**Table 2 biosensors-12-00806-t002:** Number of individuals using each antihypertensive medication class (%).

Anti-Hypertensive Agent	Number (%)
ACEI	15 (21.8)
ARB	28 (40.6)
BB	7 (10.1)
CCB	6 (8.7)
Combination of antihypertensive medication	13 (18.8)

ACEI = angiotensin-converting enzyme inhibitors, ARB = angiotensin II receptor blockers, BB = beta blockers, CCB = calcium channel blockers.

**Table 3 biosensors-12-00806-t003:** ROC parameters for the two assays performed to evaluate the expression of ENaC on platelets.

Assay	Cutoff	Area under the ROC Curve	Sensitivity (CI)	Specificity (CI)
Western blotting	0.8906	0.8331	0.7509% 0.7405–0.9256)	0.7090% (0.4779–0.8087)
AuNPs-anti-ENaC	0.5219	0.7303	0.7472% (0.656–0.8149)	0.5849% (0.5531–0.7942)

## Data Availability

Not applicable.

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
