# Peer review of "Clinical Application of Epithelial Sodium Channel (ENaC) as a Biomarker for Arterial Hypertension"

_biosensors, 2022, doi:10.3390/bios12100806_

Round 1

Reviewer 1 Report (New Reviewer)

The manuscript submitted to Biosensors by García-Rubio et al. (biosensors-1862578) mainly describes the clinical application of epithelial sodium channel (ENaC) as a biomarker of arterial hypertension. Considering the seriousness of global public health problems caused by arterial hypertension, the method established in this study can detect ENaC quickly through gold nanoparticles, which is especially suitable for initial screening and early diagnosis. It is a scientific research attempt worth encouraging. However, there are still some problems in the research described in the manuscript, which need to be further explained and improved by the authors.

1. In the Introduction section, the authors are advised to supplement the corresponding references to support their viewpoints, such as lines 29-32, 36-39, and 42-43.

2. In the Introduction section, the authors should introduce the epithelial sodium channel (ENaC) in detail, including but not limited to its tissue expression distribution, protein structure, physiological and pathological functions, etc. And explain in detail why ENaC can be used as a biomarker for the diagnosis of arterial hypertension and how its specificity is.

3. A row header appears to be missing from Table 2. In addition, Table 3 and Table 1S are not shown in the manuscript, but Table 4 is shown.

4. In line 270, it should be “ACEI”.

5. Figure 1, Figure 3 and Figure 4 are incomplete, and the clarity and pixels of these pictures need to be further improved by the authors.

6. In Figure 5, the WB band results of the house keeping protein (GAPDH) must also be provided.

7. In the Discussion part, it is necessary to compare the similarities and differences between the new detection method constructed by the authors and the methods that have been commonly used in clinic, as well as the advantages and disadvantages of the new method, and look forward to the future improvement methods and directions.

8. In supplementary material Figure 1S, the explanatory of the picture seems to be confusing. And there is no one-to-one correspondence between the description content and the picture display content. In addition, I want to know why the authors use BSA to cover gold nanoparticles.

9. The gold nanoparticles methods established by this manuscript, what are the reproductivities of these methods?

10. The references cited by the authors are too out-of-date to reflect the latest research progress, and the authors are advised to quote the up-to-date references.

Author Response

Reviewer 1

  1. In the Introduction section, the authors are advised to supplement the corresponding references to support their viewpoints, such as lines 29-32, 36-39, and 42-43.

We have included the respective references. Thank you

  1. In the Introduction section, the authors should introduce the epithelial sodium channel (ENaC) in detail, including but not limited to its tissue expression distribution, protein structure, physiological and pathological functions, etc. And explain in detail why ENaC can be used as a biomarker for the diagnosis of arterial hypertension and how its specificity is.

We have included additional information. Thank you

  1. A row header appears to be missing from Table 2. In addition, Table 3 and Table 1S are not shown in the manuscript, but Table 4 is shown.
  2. In line 270, it should be “ACEI”.

We have amended the missing information. Thank you

  1. Figure 1, Figure 3 and Figure 4 are incomplete, and the clarity and pixels of these pictures need to be further improved by the authors.
  2. In Figure 5, the WB band results of the house keeping protein (GAPDH) must also be provided.

Figures have been improved

  1. In the Discussion part, it is necessary to compare the similarities and differences between the new detection method constructed by the authors and the methods that have been commonly used in clinic, as well as the advantages and disadvantages of the new method, and look forward to the future improvement methods and directions.

This is a very pertinent commentary. We have included in the Discussion section. Thank you

  1. In supplementary material Figure 1S, the explanatory of the picture seems to be confusing. And there is no one-to-one correspondence between the description content and the picture display content. In addition, I want to know why the authors use BSA to cover gold nanoparticles.

We have corrected the figure legends. Thank you

  1. The gold nanoparticles methods established by this manuscript, what are the reproductivities of these methods?

 It is a very good question. We have included a paragraph in page 21 line 589-599 Discussion section referring to the reproducibility of AuNPs assays.

  1. The references cited by the authors are too out-of-date to reflect the latest research progress, and the authors are advised to quote the up-to-date references.

The references have been updated. Thank you

Reviewer 2 Report (New Reviewer)

The authors of the manuscript ‘Clinical application of epithelial sodium channel (ENaC) as a biomarker for arterial hypertension’ explored that AuNP-anti-ENaC assay could be used for early diagnosis to critically improve opportune clinical management of HTN.

There are several questions as below about this manuscript:

1. The detail method of Principal Components Analysis (PCA) should be provided, e.g. the version of R package used in the manuscript.

2. In Figure 5, I wonder that is there significant difference between the controls and patients by using Western blotting or AuNPs-anti-ENaC assay. In addition, the AuNPs-antiENaC assay should write as AuNPs-anti-ENaC assay.

3. The authors should provide the figure of ROC.

4. The representative GAPDH figure should be provided in Figure 5.

5. Please check whole manuscript carefully since there are some format errors.

e.g. There is an extra full stop in the ‘85% of the variation in the data. . ’ in the figure legends 1 and 2.

Author Response

Reviewer 2

There are several questions as below about this manuscript:

  1. The detail method of Principal Components Analysis (PCA) should be provided, e.g. the version of R package used in the manuscript.

We have completed the information. Thank you

  1. In Figure 5, I wonder that is there significant difference between the controls and patients by using Western blotting or AuNPs-anti-ENaC assay. In addition, the AuNPs-antiENaC assay should write as AuNPs-anti-ENaC assay.
  2. The authors should provide the figure of ROC.
  3. The representative GAPDH figure should be provided in Figure 5.

 All these observations were very pertinent. We have improved Figure 5. Thank you

  1. Please check whole manuscript carefully since there are some format errors   e.g. There is an extra full stop in the ‘85% of the variation in the data. . ’ in the figure      legends 1 and 2.

Thank you, we have corrected the format errors.

Reviewer 3 Report (New Reviewer)

Gold nanoparticles are more and more widely used not only in the technology industry, everyday use products, but also in the biomedical industry. In the latter case, all new diagnostic tests are an important element that can be used for the initial screening of various diseases.

Hypertension is a very big problem in terms of the health of the public age of countries. It is also a very important risk factor for cardiovascular disease and kidney failure.

The authors in the manuscript presented for review entitled "Clinical application of epithelial sodium channel (ENaC) as a biomarker for arterial hypertension" described the possibilities of using a diagnostic test for the early detection of hypertension in which gold nanoparticles conjugated with the ENaC antibody expressed on blood platelets were used.

The studied group of patients (n = 167) is, in my opinion, sufficient to conclude preliminary conclusions resulting from the above-mentioned studies (the cut-off value for the AuNP-anti-ENaC test was confirmed in 62.1% of patients with clinical hypertension and it was additionally demonstrated in 59.7% patients had previously undiagnosed hypertension).

The description of the test group and the method of obtaining gold nanoparticles is correct.

In my opinion, the manuscript should be supplemented with information about the enlargement of TEM photos or additionally put photos from high magnification bioconjugates and platelet.

Yours faithfully,  

Author Response

Reviewer 3

  1. In my opinion, the manuscript should be supplemented with information about the enlargement of TEM photos or additionally put photos from high magnification bioconjugates and platelet. 

We have included clearer high magnification photos of bioconjugates (AuNPs and platelets). Thank you

This manuscript is a resubmission of an earlier submission. The following is a list of the peer review reports and author responses from that submission.

Round 1

Reviewer 1 Report

Although the authors were made effort  to improve their manuscript this is not enough. 

I have to suggest rejection.

Reviewer 2 Report

The paper "Clinical application of ephitelial sodium channel ..." by Garcia-Rubio et al. deals out an investigation on the application of a hypertension diagnostic assay based on gold nanoparticles conjugated to an antibody against ephitelial sodium channel. The authors have previously reported  that overexpression of ENaC on the plasma membrane of human platelets is a hallmark of arterial hypertension. The authors show that the new diagnostic assay is able to confirm the diagnosis for 62.1% of patients with clinical arterial hypertension and detected 59.7% of pattients with previously undiagnosed arterial hyprtension. The authors correctly report that this non-high accuracy of their diagnostic assay needs to be improved and their application of antibody conjugate gold nanoparticles  does indeed appear to be at early stage.

I suggest a major revision of the paper.

The presentation is confused and the entire paper seems to be hastily written. See the last (no-sense) sentence at the Conclusions section, as for example. A part various typos and the continuous use of "gold nanoparticles" instead of the corresponding acronym AuNPs  introduced in the abstract, a part such examples, the paper presents many ripetitions and some confused sentences.

The caption of figure 1 is uncorrect and misleading, only SEM e TEM images are reported.

In sample size calculation, what is W2?

trhe results inherent J-Values should be commented.

The suggestion to use AuNP-antiENaC assay as a screening test seems to be  premature in the light of the current results obtained, and I do not understand on what basis the authors advance this suggestion.  A diagnostic AuNP-antiENaC assay could become an arterial hypertension screening test if their usage is improved in term of accuracy, specificity and sensitivity.

Reviewer 3 Report

This work reports an immunoassay based on AuNP-anti-ENaC conjugates for the diagnosis of arterial hypertension. However, the authors have recently published a very similar paper on Biosensors and Bioelectronics, 157 (2020) 112151. Therefore, rejection is recommended for duplicate submissions.